# Linking Circulating Serum Proteins with Clinical Outcomes in Esophageal Adenocarcinoma—An Emerging Role for Chemokines

**DOI:** 10.3390/cancers12113356

**Published:** 2020-11-13

**Authors:** Noel E. Donlon, Andrew Sheppard, Maria Davern, Fiona O’Connell, James J. Phelan, Robert Power, Timothy Nugent, Kate Dinneen, John Aird, John Greene, Paul Nevins Selvadurai, Anshul Bhardwaj, Emma K. Foley, Narayanasamy Ravi, Claire L. Donohoe, John V. Reynolds, Joanne Lysaght, Jacintha O’Sullivan, Margaret R. Dunne

**Affiliations:** 1Department of Surgery, Trinity Translational Medicine Institute, Trinity College Dublin, St James’s Hospital, Dublin 8, Ireland; donlonn@tcd.ie (N.E.D.); SHEPPAA@tcd.ie (A.S.); DAVERNMA@tcd.ie (M.D.); OCONNEFI@tcd.ie (F.O.); PHELANJ1@tcd.ie (J.J.P.); POWERR8@tcd.ie (R.P.); NUGENTTI@tcd.ie (T.N.); Anshul.Bhardwaj@tcd.ie (A.B.); EmmFoley@stjames.ie (E.K.F.); nravi@stjames.ie (N.R.); claire.donohoe@tcd.ie (C.L.D.); reynoljv@tcd.ie (J.V.R.); jlysaght@tcd.ie (J.L.); osullij4@tcd.ie (J.O.); 2Trinity St James’s Cancer Institute, St James’s Hospital, Dublin 8, Ireland; 3Department of Histopathology, St James’s Hospital, Dublin 8, Ireland; katedinneen4@gmail.com (K.D.); johnaird@mater.ie (J.A.); 4Department of Medical Oncology, St James’s Hospital, Dublin 8, Ireland; greenejo@tcd.ie (J.G.); paulnevinsselvadurai@svhg.ie (P.N.S.)

**Keywords:** esophageal adenocarcinoma, serum markers, cytokines, chemokines, clinical outcome, survival, treatment response, prognostic markers

## Abstract

**Simple Summary:**

Cancer of the esophagus (food pipe) is an aggressive type of cancer with poor prognosis and rates are increasing. Current treatments help to prolong survival but only for a minority of patients, therefore there is an urgent need to discover why some people do not respond and to develop new and improved treatments. Newer treatments targeting the immune system show promise but the anti-tumor immune response in esophageal cancer is not well understood. This study measured levels of 54 immune markers in serum of patients with esophageal cancer and evaluated a link with patient clinical outcomes, e.g., survival time, response to treatment, and adverse events. We found that certain chemokines, proteins which control immune cell trafficking, were particularly high in patients who survived longer (CCL22 and CCL26) and responded to treatment (CCL4), suggesting the importance of immune cell movement in orchestrating an effective immune response to esophageal cancer.

**Abstract:**

Esophageal adenocarcinoma (EAC) is an aggressive cancer with poor prognosis and incidence is increasing rapidly in the Western world. Multi-modal treatment has improved survival outcomes but only for a minority of patients. Currently no markers have been identified to predict treatment response. This study investigated the association between clinical outcomes and pre-treatment levels of 54 serum proteins in n = 80 patients with EAC. Low tumor regression grade (TRG), corresponding to a favorable treatment response, was linked to prolonged overall survival (OS). CCL4 was higher in patients with a favorable treatment response, while Tie2 and CRP were higher in poor responders. Elevated CCL22 and CCL26 was associated with improved OS, while elevated IL-10 showed a negative association. CCL3, CCL4, IL-1α and IL-12/IL23p40 were highest in individuals with no adverse features of tumor biology, whereas levels of Tie2 and VEGF were lowest in this cohort. CCL4 was also elevated in patients with high tumor lymphocyte infiltration. Comparison of matched pre- and post-treatment serum (n = 28) showed a large reduction in VEGFC, and a concomitant increase in other cytokines, including CCL4. These data link several serum markers with clinical outcomes, highlighting an important role for immune cell trafficking in the EAC antitumor immune response.

## 1. Introduction

Adenocarcinoma of the esophagus (EAC) or the esophagogastric junction (EGJ) are particularly aggressive cancer types and are rapidly increasing in the Western world in line with growing obesity rates [1]. In locally advanced disease, the addition of neoadjuvant chemoradiotherapy or perioperative chemotherapy (neo-CT) to surgical resection provides a modest overall survival (OS) benefit [2,3,4]. This benefit is mainly restricted to the 10–15% of patients who display a pathological complete response (pCR) to treatment in the resected tumor [5]. As prognosis is poor [6] and systemic therapy is associated with significant morbidity [7], the ability to predict a response to neoadjuvant treatment could allow a more tailored approach to multimodal therapy.

The pathological response to neo-CT treatment can be measured by the tumor regression grade (TRG) [8]. Use of the Mandard TRG classification system, first described in 1994 [9], is recommended by UK and Irish guidelines [10]. This system is based on the amount of residual tumor and the degree of fibrosis at the primary tumor site and uses a five-point scale, where a score of one corresponds to complete tumor regression with no tumor cells detectable and a score of five denotes no evidence of tumor regression. A pathological response to neo-CT, as defined by a Mandard TRG score of 1–2, is associated with prolonged disease-free survival (DFS) and OS in retrospective studies [11,12,13] and prolonged OS in a secondary analysis of a randomized trial [14]. Other morphological markers, including downstaging of lymph node status, present strong independent prognostic markers in EAC [15], particularly in patients without a local pathological response [14]. Adverse pathological features described on resected tumor samples, including poor differentiation, mucinous or signet ring features, and perineural, vascular and lymphovascular invasion are also negatively associated with DFS [16,17,18,19] and have the potential to be used alongside TRG in patient stratification.

Activation of the effector immune response is now recognized to be involved in the response to chemoradiotherapy [20]. Immunogenic cell death is a key mediator of this and can be induced by radiotherapy and cytotoxic chemotherapies employed in the neoadjuvant setting [21,22]. This therapy-induced anti-tumor immunity may predict tumor regression, for example a tumor gene expression signature of the DNA damage immune response, including programmed cell death ligand 1 (PD-L1) and several inflammatory cytokines (CXCL9, CXCL13, CXCL10/IP-10, CXCL11, CCL5, CCL18), has been shown to predict tumor regression in EAC [23]. Interestingly, expression of such DNA damage immune response markers was also associated with increased levels of CD8^+^ tumor-infiltrating lymphocytes in resected tumors, which have independently been associated with pathologic response in EAC [24]. However, the association between pathological responses and blood-derived immune profiles has not been previously explored in detail in EAC. Chronic inflammation has been implicated in EAC development and presents a possible mechanism of resistance to neoadjuvant therapy [25]. Circulating proteins can be quantified easily as a measure of systemic events such as inflammation and angiogenesis and are more amenable to routine interrogation than tumor tissues. Circulating factors such as C-reactive protein (CRP) [26], soluble interleukin-6 receptor (sIL-6R) [27] and vascular endothelial growth factor (VEGF) [28] have demonstrated prognostic potential in esophageal cancer, however most studies to date have focused on squamous cell type esophageal tumors. We aimed to assess whether such serum protein profiles also had a prognostic or predictive ability in the EAC setting.

This study quantified levels of 54 serum markers from EAC/EGJ patients, and assessed a link with clinicopathological outcomes, e.g., overall survival (OS), TRG, adverse events and immune cell infiltration into tumors. Through this, we aimed to better understand systemic immune and angiogenic profiles linked with clinical outcomes.

## 2. Results

### 2.1. Lower TRG Scores and Node Negativity Are Associated with Longer Overall Survival Time 

OS time was visualized using a Kaplan–Meier curve, dividing patients into low (TRG1–2) and high (TRG3–5) groups or based on treatment response and nodal status, as shown in Figure 1. Log-rank testing showed that low TRG scores were associated with a longer survival time (median 70.4 months) compared to high scores (median 33.5 months), *p* < 0.05 (Figure 1a). EAC patients with node positive disease showed shorter survival times (median 22.4 months) than node negative (94.6 months) or pathological complete response (pCR) (70.1 months) cohorts (Figure 1b).

### 2.2. CCL4 Is Lower and Tie2 and CRP Levels Higher in Pre-Treatment Serum of Patients with a Subsequent Poor Response to Neo-Adjuvant Treatment

Given the considerable survival differences observed between low (TRG1–2) and high (TRG3–5) groups, these divisions were used to assess differences in the levels of serum cytokines collected at a pre-treatment timepoint (Figure 2). Patients with a favorable response to neo-CT, i.e., a low TRG score, showed higher levels of circulating CCL4 (*p* < 0.01) (Figure 2a). A further subdivision of TRG groups showed that levels were higher in the TRG1/2 cohort when compared to TRG3 (*p* < 0.05) and TRG4/5 (*p* < 0.01) groups individually (Figure 2b). Levels of angiogenic factor Tie2 (*p* < 0.05) and inflammatory factor CRP (*p* < 0.05) were higher in poor responders however (Figure 2c–f). For Tie2, this difference was most apparent when TRG1/2 and TRG4/5 groups were compared. The remaining proteins measured did not show any significant differences between TRG groups (Appendix A).

### 2.3. Pre-Treatment Serum IL-10 Is Associated with Reduced OS, while CCL22 and CCL26 Are Associated with Prolonged OS

After observing that levels of certain serum proteins measured at a pre-treatment timepoint were linked with TRG scores, we next investigated whether protein levels were also associated with OS time. Patients were divided into high and low expressing cohorts based on median expression value of each protein, and associations were assessed by log-rank test and visualized using Kaplan–Meier curves (Figure 3). As shown in Figure 3, higher than median levels of the immunosuppressive cytokine IL-10 were associated with poorer OS (*p* = 0.0055, hazard ratio (HR) = 0.378, 95% confidence interval (CI) = 0.1818–0.786), whereas higher than median levels of the chemokines CCL22 (*p* = 0.0101, HR = 2.301, 95% CI = 1.183–4.475) and CCL26 (*p* = 0.0163, HR = 2.254, 95% CI = 1.14–4.456) were associated with longer OS. The remaining proteins measured showed no significant association with OS time (Appendix A). No significant differences were observed when cytokine levels were assessed between patients separated into pathological node negative and node positive groups (Appendix A), or recurrence versus non-recurrence groups (Appendix A).

### 2.4. Reduced Levels of Circulating CCL3, CCL4, IL-1α and IL-12/IL-23p40 and Elevated Levels of Tie2 and VEGF Are Associated with Adverse Tumor Features

Recent work by our group has shown that adverse features of tumor biology confer reduced survival rates in esophageal cancer [29]. We therefore sought to investigate differences in the levels of circulating factors in the serum of patients with or without adverse features. Adverse pathological features included poor tumor differentiation, mucinous or signet ring features, and evidence of perineural, vascular and lymphovascular invasion. Patients were divided into three cohorts; 0, 1–2 and 3–4 adverse features. As shown in Figure 4, levels of CCL3, CCL4, IL-1α, and IL-12/IL-23p40 were observed to be highest in those with no adverse features compared to the groupings with adverse features. Conversely, levels of Tie2 and VEGF were lowest in those with no adverse features present.

### 2.5. Neo-Adjuvant Treatment Increases Serum Pro-Inflammatory Cytokines and Decreases Anti-Angiogenic Mediators

Given extensive evidence in the literature outlining the various pro- and anti-tumor effects of chemotherapy and radiotherapy on anti-tumor immunity, we compared the expression of circulating serum levels of immune-based proteins before and after neo-CT treatment in order to further elucidate the effect of neo-adjuvant treatment on systemic immunity. Serum specimens were collected at a pre- and post-treatment timepoint for n = 28 donors (11 donors received CROSS regimen, 7 received FLOT and 10 received MAGIC), and levels of 54 circulating proteins were quantified on the same plate, within a single assay run. As shown in Figure 5, significant elevations were observed after treatment in the levels of PlGF (*p* < 0.0001), CCL3, IL-21, IL-12/IL-23p40, GM-CSF, bFGF, TNF-α, IFN-γ, CXCL10, CCL4, and IL-5, whereas angiogenic factor VEGFC was decreased (*p* < 0.0001) when data were analyzed using a Wilcoxon matched pairs signed rank test. Such changes remained evident for the majority of cytokines when the cohort was divided up by treatment type–CROSS, FLOT or MAGIC (Appendix A).

### 2.6. Tumors with High Lymphocytic Infiltration Showed Higher Levels of Circulating TNF-β, CCL4, CCL13 and IL-27

Hematoxylin and eosin stained tumor pre-treatment biopsy slides were available for a subset of n = 32 patients who underwent serum cytokine analysis. Within this cohort, consensus scoring from two pathologists showed an overall low level of tumor stroma (only 3/32 tumors scored >50%) and high levels of overall inflammation (29/32 donors scoring highly). Infiltration of lymphocytes, plasma cells, neutrophils, and eosinophils was also classified into high (>50%) and low (0–50%) groups. When patients were grouped into lymphocyte high (>50% infiltration) and low (<50%) cohorts, it was observed that individuals with lymphocyte high tumors also showed elevated levels of serum CCL4 (*p* > 0.05), CCL13 (*p* > 0.05), IL-27 (*p* > 0.05), and TNF-β (*p* > 0.01) (Figure 6). No significant differences were observed when levels were compared for other cytokines (Appendix A), or with infiltration of other immune cell types, i.e., plasma cells, eosinophils, or neutrophils.

### 2.7. Correlation Analysis of Cytokines with Patient Clinical Outcomes

To determine if treatment-naïve protein expression levels identified via 54-plex ELISA correlated with patient clinical features, Spearman correlations were performed and visualized using the R package “CorrPlot” (Figure 7). Nine cytokines with fewer than 40 detectable readings (i.e., half the total cohort tested) were excluded from analysis. The corresponding Spearman r values and *p* values associated with each correlation are summarized in Figure 7b. Age at time of diagnosis indicated weak significant positive correlations with secretion of IL-12/IL-23p40 (r = 0.2702, *p* = 0.015), IL-6 (r = 0.3404, *p* = 0.002), CCL3 (r = 0.2622, *p* = 0.02), PlGF (r = 0.2902, *p* = 0.009), and TNF-α (r = 0.2387, *p* = 0.032). Perineural invasion showed a weak negative correlation with CXCL10 (r = −0.268, *p* = 0.016). Lymphatic invasion showed weak positive correlations with IL-1RA (r = 0.2363, *p* = 0.035) and IL-27 (r = 0.2298, *p* = 0.04) and serosal involvement showed weak negative correlations with CCL4 (r = −0.2513, *p* = 0.027) and VEGF (r = −0.2822, *p* = 0.013). Clinical TNM stage showed weak positive correlations with IFN-γ (r = 0.3229, *p* = 0.024) and CCL2 (r = 0.3274, *p* = 0.017), while pathologic T stage yielded weak negative correlations with CXCL10 (r = −0.223, *p* = 0.048), CCL4 (r = −0.2793, *p* = 0.013), and PlGF (r = −0.2767, *p* = 0.014), with pathologic N stage showing weak negative correlations with CCL11 (r = −0.2234, *p* = 0.048), IL-22 (r = −0.2517, *p* = 0.029), and positive weak correlations with IL-17B (r = 0.2554, *p* = 0.027). TRG indicated weak negative correlations with CCL4 (r = −0.2694, *p* = 0.016) and CCL17 (r = −0.2257, *p* = 0.046), with disease recurrence showing weak positive correlation with IL-1β (r = 0.2923, *p* = 0.027), also yielding weak negative correlations with CCL17 (r = −0.2503, *p* = 0.026) and TNF-β (r = −0.2323, *p* = 0.046). No other significant associations were identified between treatment naive protein expression and matched clinical features.

## 3. Discussion

TRG scoring upon surgical resection represents a promising candidate for stratifying patients into good and poor prognostic groups after neo-CT treatment. Previous multicenter cohort studies have demonstrated the independent prognostic value of the Mandard classification of TRG for both neoadjuvant chemotherapy and chemoradiotherapy [11,12,13]. We observed here in a cohort of n = 80 patients, that those with a low TRG score have a longer survival compared to those with high TRG. Previous studies on chemoradiotherapy-treated patients found that a three-point scale (TRG 1 vs. TRG 2–3 vs. TRG 4–5) provided the best discriminant fit [30], while a more recent large (n = 1,293) multicenter analysis of neoadjuvant chemotherapy concluded that separating patients into TRG 1–2 and TRG 3–5 had a greater prognostic value [15]. We also observed that splitting TRG cohorts in the latter, two-tiered manner gave the strongest prognostic result and therefore used this approach throughout this study. While the TRG score appears to have a consistent prognostic ability, there is an urgent clinical need to implement non-invasive stratification approaches into routine clinical practice at the pre-treatment stage in order to identify patients who are likely to benefit from neoadjuvant treatment and those who are not. Patients in the non-responsive group could then proceed immediately to surgical intervention or alternative treatment approaches and would avoid undesirable side effects and delays to surgery, which often leads to tumor upstaging and worse prognosis in those who fail to benefit from neoadjuvant treatment. We therefore investigated levels of 54 circulating serum cytokines, in order to assess their potential association with clinical outcomes.

Analysis of 54 markers in treatment-naive serum showed associations between several circulating factors and clinical outcomes. Of note, chemokines CCL22, CCL26, and CCL4 were all associated with favorable outcomes, with elevated CCL22 and CCL26 linked with longer OS time and higher CCL4 observed in patients who responded well to neoadjuvant therapy (TRG 1–2). CCL4 was particularly prominent throughout our analysis, being also observed at the highest levels in patients with no adverse features (alongside CCL3, IL-1α, and IL-12/IL-23p40), elevated in patients with high tumor lymphocyte infiltration and negatively associated with serosal invasion and pathologic tumor stage. In terms of adverse clinical associations, IL-10 was higher in the serum of patients with a shorter survival time, and Tie2 and CRP were higher in pre-treatment serum of patients who had a subsequent poor response to neo-CT therapy. The angiogenic factors Tie2 and VEGF were also elevated in the serum of patients with adverse features.

Chemokines form part of a complex network of chemotactic molecules, responsible for orchestrating immune cell migration and infiltration into tissues and display high levels of redundancy. Levels of various chemokines have previously been associated with clinical outcomes including OS and tumor recurrence, but whether such associations are favorable or adverse appears to depend on the tumor type and whether the chemokine is measured in the local tumor environment or in circulation [31]. Prognostic analysis of data from The Human Protein Atlas showed that CCL22 expression in tumors was favorably associated with survival in colorectal, endometrial cancers, and in head and neck cancers [31]. Elevated serum CCL22 was also linked with improved survival in a cohort of 1208 patients with glioma [32]; however, serum CCL22 was found to be elevated in advanced tumor stages in breast cancer [33] and was associated with metastatic spread and recurrence in gastric cancer patients [34]. We observed that higher than median levels of serum CCL22 were associated with prolonged OS in the EAC/EGJ setting (*p* = 0.0101, HR = 2.301, 95% CI = 1.18–4.48). CCL22 is produced by macrophages, dendritic cells and tumor cells and is a chemoattractant for several cell types including regulatory T cells and T helper type 2 cells, as evidenced by their expression of the CCL22 receptor, CCR4 [35]. Despite the immunoregulatory, a potential tumor-promoting action ascribed to such effector T cells, the presence of elevated systemic CCL22 may also indicate uninhibited function of antigen presenting cells, which are important for orchestrating anti-tumor responses and therefore result in a net positive effect when elevated in the periphery. Indeed, previous work by our colleagues has demonstrated that CCL22 expression in EAC tumors is relatively low compared to EAC serum [36]. Much less is known about the role of CCL26 in cancer, however elevated levels of this chemokine in tumors have been reported in later stage colorectal cancer and has been associated with worse prognosis [37]. We observed that higher than median levels of serum CCL26 were associated with prolonged overall survival (*p* = 0.0163, HR = 2.254, 95% CI = 1.14–4.45). CCL26 is responsible for recruitment of eosinophils to the mucosa, and is strongly upregulated in the chronic inflammatory condition, eosinophilic esophagitis [38] and therefore, its abundance in circulation in the EAC setting could also reflect advantageous mucosal inflammation.

CCL4, also known as macrophage inflammatory protein-1β (MIP-1β) is a CC chemokine expressed by various immune cells, with specificity for CCR5 receptors. Elevated expression of CCL4 in tumors was associated with unfavorable prognosis in renal carcinoma, but favorable prognosis when detected in colorectal, endometrial, and melanoma tumors [31] and esophageal squamous cell carcinoma [39]. In the latter study, expression of CCL4 in tumors was associated with higher cytotoxic T cell infiltration and expression of cytolytic effector molecules perforin and granzyme [39]. Interestingly, we observed that patients with high lymphocytic tumor infiltration showed elevated serum CCL4 (*p* < 0.05), when compared to those with low infiltration. Although we observed no significant association between serum CCL4 and OS, we observed higher CCL4 levels in patients with a good treatment response (*p* < 0.01), those with no adverse features, and found a negative association with serosal invasion and pathologic tumor stage, implicating CCL4 as a marker of favorable anti-tumor immune responses in EAC. CCL4 expression in serum was observed to be higher in EAC patients (n = 14) compared to squamous cell tumors (n = 36) in one study, suggesting potential differences in the immune response to these histological cancer subtypes [40].

Conflicting studies on the prognostic potential of chemokines highlight the stark differences in the prognostic value between different cancer types and measurement locations, perhaps reflecting a context-dependent role of the immune cells these chemokines recruit to tumors. The strong male predominance in our EAC cohort may also add a further layer of complexity to these observations, which needs to be addressed in future studies. Growing evidence suggests that chemokines represent useful biomarkers in the esophageal cancer setting [39]. However, given the high level of redundancy and complexity of chemokines, it is likely that profiles composed of multiple chemokines will prove more prognostically useful than any single player alone. Indeed, a study on esophageal squamous cell carcinoma showed that elevated tumor expression of CCL4 predicted prolonged survival, but a CCL4^high^/CCL20^low^ group demonstrated better overall survival, whereas CCL4^low^/CCL20^low^ and CCL4^low^/CCL20^high^ groups showed the worst overall survival [41]. Although we did not observe any link between CCL4, CCL20, and OS in our EAC cohort (even when analyzed in combination), we did however observe improved OS time when favorable cytokine profiles (CCL22^high^, CCL26^high^, and IL-10^low^) were analyzed in combination, reinforcing the idea that multiple markers enhance prognostic effects.

In terms of markers with negative clinical associations, levels of pre-treatment serum CRP and Tie2 were increased in patients with a subsequent poor treatment response to neoadjuvant therapy. CRP is an acute phase protein and general indicator of inflammation and has previously been linked with poor prognosis, recurrence and response to therapy in several malignancies, including EAC [42,43]. Such observations led to development of the modified Glasgow Prognostic Score, which combines indices of decreased plasma albumin and elevated CRP. Although this scoring approach has been shown to have prognostic ability in other gastrointestinal cancer types [44], a prognostic effect has not been demonstrated in esophageal cancer, where an association with advanced stage but no independent prognostic significance or impact on operative outcomes was observed in a cohort of 223 patients [45]. We did not observe any association between CRP levels and OS in our cohort, possibly due to the cohort size (n = 80), which was smaller than that of the studies of Ikeda et al. (n = 356) [42] and Nozoe et al. (n = 262) [43]. However, CRP was observed to be higher in patients with a poor response to neo-CT, i.e., TRG3–5, suggesting a negative role for systemic inflammation in the treatment response in this cohort.

We also observed that patients with a poor treatment response had elevated pre-treatment levels of Tie2, a tyrosine kinase that acts as a cell surface receptor for angiopoietins and plays an important role in angiogenesis. Circulating Tie2 is a vascular response biomarker in bevacizumab-treated ovarian and metastatic colorectal cancer patients [46]. The angiopoietin/Tie2 axis is therefore an attractive target for cancer therapy, since it is involved in inflammation, metastasis and lymphangiogenesis, and drugs such as trebananib have been designed to block Tie2 receptor interaction with angiopoeitin ligands [47]. We observed that patients with adverse features showed a higher pre-treatment level of angiogenic factors Tie2 and VEGF, however these cytokines showed no significant difference when other clinical parameters were assessed. A higher than median expression of the immunosuppressive cytokine IL-10 in treatment-naive serum was associated with a shortened survival time, in agreement with other studies in gastrointestinal cancers and a meta-analysis of 21 studies on 1788 patients with various cancer types, which showed that high serum IL-10 was associated with worse clinical outcomes [48].

Proinflammatory cytokines IL-1α and IL-12/IL-23p40 and chemoattractant molecules CCL3 and CCL4 were detected at lower levels in the treatment-naive serum of patients with adverse tumor features when compared to those with no adverse tumor features, potentially indicating a lack of an effective anti-tumor response, and instead reflecting an immune environment which promotes disease progression and the acquisition of adverse tumor features. Alternatively, the presence of adverse tumor features may reflect a more aggressive and immunosuppressive tumor microenvironment which actively dampens anti-tumor immunity. IL-1 acts downstream of inflammasome signaling and as such plays a vital role in innate immunity and can act as a damage associated molecular pattern (DAMP) [49]. Conversely, significantly higher levels of tumor-promoting, pro-angiogenic factors Tie2 and VEGF were detected in the pre-treatment serum of patients who had adverse tumor features following neo-adjuvant treatment which are associated with poorer responses and prognoses [29].

Chemotherapy and radiotherapy can have profound effects on reshaping the tumor microenvironment and immune response to cancer, leading to either an enhancement or suppression of anti-tumor immune responses [21,50], which depend on a myriad of tumor–host interactions. Therefore, we investigated levels of immune serum proteins before and after neo-CT treatment to help elucidate the net effect of neoadjuvant therapy on systemic immunity. Of the 54 mediators measured, twelve showed significant alterations following neo-CT treatment, including: VEGFC, PlGF, CCL3, IL-21, IL-12/IL-23p40, GM-CSF, bFGF and TNF-α, IFN-γ, CXCL10, CCL4 and IL-5, with VEGFC being lower after treatment, whereas all other effectors were elevated at the post-treatment, pre-surgical time point. Such factors comprise a mixture of pro- and anti-tumor immune effector molecules, highlighting the double-edged sword effects of chemoradiotherapy. Interestingly, while we did not observe any significant changes in levels of VEGF family members, in VEGFA and VEGFD after neo-CT treatment, in line with previous studies [51], levels of the family member VEGFC were greatly reduced after neo-CT treatment. Chemoradiotherapy concomitantly induces damage to tumor cells as well as the tumor vasculature [52], therefore an increase in systemic vascular damage proteins and pro-survival growth factors including bFGF and PlGF following neoadjuvant treatment is logical. bFGF has a myriad of cellular functions, including proliferation, cell survival, differentiation, and motility and has specific roles in wound healing and tissue repair, which may have adverse implications for tumor progression [53]. Esophageal cancer studies have demonstrated that FGF2 overexpression is associated with a risk of recurrence of disease as well as reduced OS post-surgical resection [54].

In summary, the data show that non-invasive serum cytokine signatures differ based on treatment response as determined by Mandard TRG score, overall survival time, adverse tumor features, and immune cell infiltration into tumors. In particular, we show that several chemokines are linked with favorable outcomes, including OS (CCL22, CCL26) and treatment response (CCL4), implicating a role for immune cell trafficking in the immune response to EAC. CCL4 in particular was linked with several favorable clinical indicators, but not OS, highlighting a need for further study to understand the underlying biology of this chemokine in EAC. A better understanding of immune signatures associated with favorable clinical outcomes in EAC will help build a clearer picture of the critical pathways involved in anti-tumor immunity and may even offer new insights on how to improve response to immune checkpoint inhibitors.

## 4. Materials and Methods

### 4.1. Ethics Statement

Ethical approval was granted from Tallaght/St James’s Hospital committee (reference 2011/27/01) and all sample and data collection was carried out using the best clinical practice guidelines. All procedures followed were in accordance with the ethical standards of the responsible committee on human experimentation (institutional and national) and with the Helsinki Declaration of 1975, as revised in 2008.

### 4.2. Patient Cohort

A cohort of 80 patients with locally advanced EAC or EGJ cancers were included for clinical analysis and these underwent serum analysis upon giving informed consent (see Table 1). These cohorts received pre-operative chemoradiation (cisplatin, fluorouracil and radiation therapy (40–44 Gy)) or the CROSS protocol of carboplatin, paclitaxel, and 41.6 Gy irradiation. Patients not suitable for CRT received pre- and post-operative chemotherapy as per the MAGIC trial regimen using etoposide, cisplatin or oxaliplatin, and fluorouracil or capecitabine and in contemporaneous terms, as per the FLOT regimen consisting of docetaxel, oxaliplatin, leucovirin, and 5-fluorouracil. The group was evaluated based on Mandard TRG status applied throughout, with TRG 1 indicating no residual cancer cells, TRG 2 indicating rare residual cells, TRG 3 representing an increase in the number of residual cancer cells, but fibrosis still predominates, TRG 4 where cancer outgrows fibrosis, and TRG 5 represents a complete absence of regressive changes. Adverse tumor biological features were defined as poor differentiation, the presence of lymphatic invasion, vascular invasion, and perineural invasion in the resected specimen.

### 4.3. Quantification of Serum Immune Proteins

Serum samples from n = 80 EAC patients were collected, snap-frozen, and cryopreserved at −80 °C until analysis. Serum proteins were quantified using a V-PLEX Human Biomarker 54-plex enzyme-linked immunosorbent assay (ELISA) kit (Meso Scale Diagnostics, MD, USA), spread across 7 discrete assays. These assays quantified the secretions of the following 54 proteins: CRP, CCL11 (eotaxin), CCL26 (eotaxin-3), FGF (basic), GM-CSF, ICAM-1, IFN-γ, IL-10, IL-12/IL-23p40, IL-12p70, IL-13, IL-15, IL-16, IL-17A, IL-17A/F, IL-17B, IL-17C, IL-17D, IL-1RA, IL-1α, IL-1β, IL-2, IL-21, IL-22, IL-23, IL-27, IL-3, IL-31, IL-4, IL-5, IL-6, IL-7, IL-8, IL-8, IL-9, CXCL10 (IP-10), CCL2 (MCP-1), CCL13 (MCP-4), CCL22 (MDC), CCL3 (MIP-1α), CCL4 (MIP-1β), CCL20 (MIP-3α), PlGF, SAA, CCL17 (TARC), Tie2, TNF-α, TNF-β, TSLP, VCAM-1, VEGF-A, VEGF-C, VEGF-D and VEGFR-1/Flt-1. All assays were run as per the manufacturer’s recommendations, with an alternative protocol of overnight serum incubation being used for all assays except Vascular Injury and Angiogenesis, which were run in a single day. All results were reported in pg/mL. Protein concentrations were calculated using Meso Scale Diagnostics Discovery Workbench software (version 4.0). Values outside the kit detection limits were not reported.

### 4.4. Histological Assessment of Matched EAC Donor Tissues

Routine hematoxylin and eosin stained sections from diagnostic biopsy material from n = 32 serum-matched EAC donors were reviewed by two pathologists (JA and KD), who were blinded to the clinical outcomes. Inflammatory cell density and tumor stroma percentage were assessed in tissue fragments containing invasive carcinoma. Inflammatory cell density was classified as either absent/low-grade (mild/patchy increase in inflammatory cells) or high-grade (prominent inflammatory infiltrate and/or involvement and destruction of cancer cell islands). The presence of eosinophils, neutrophils, lymphocytes, and plasma cells was also assessed and similarly classified as either absent/low-grade or high-grade. The tumor stroma percentage (TSP) was assessed by estimating the proportion of stroma as a percentage of the visible field from an area of carcinoma, excluding areas of mucin deposition or necrosis. Tumors were classified as low TSP if stroma accounted for <50% of the visible field or high-TSP, if stroma accounted for 50% or more of the visible field.

### 4.5. Statistical Analyses

Analyses were performed using SPSS^®^ (version 18.0) software (SPSS, Chicago, IL, USA) and Prism GraphPad version 6.0. A significance level of *p* < 0.05 was used for all analyses and all the *p* values were reported are two-tailed. Continuous variables were compared using non-parametric unpaired t-tests (for two groups) or the Kruskal–Wallis test (for more than two groups) and multiple comparisons were accounted for using the Dunn’s post-hoc test. Association of categorical variables (differences for dichotomous variables between groups) was assessed using χ^2^ test. The Kaplan–Meier method and the log rank test were used to assess the differences in survival between groups. Survival time was measured from the date of the first treatment to the date of death or last follow-up. Spearman correlations were carried out using R software, version 3.6.2. Correlations were generated using R package “Hmisc”, version 4.4-0. Graphical representations of correlations were generated with the R package “CorrPlot”, version 0.84 [55]. Details of specific statistical tests used are given in each figure legend.

## 5. Conclusions

This study quantified levels of 54 serum proteins in a cohort EAC patients and showed that certain proteins, in particular chemokines, were associated with favorable clinical outcomes, including OS (CCL22, CCL26) and response to neo-CT treatment (CCL4). This suggests that immune cell trafficking plays an important role in the orchestration of an effective anti-tumor immune response in the EAC setting. Such prognostic and predictive data will be useful in guiding future studies exploring the migration of effector immune cells to EAC tumors.

## Figures and Tables

**Figure 1 cancers-12-03356-f001:**
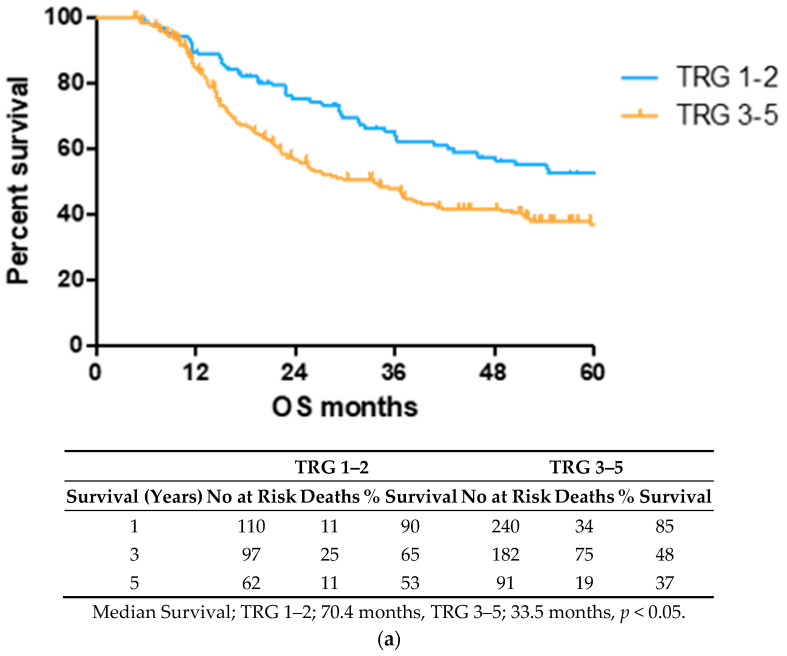
Lower TRG scores and node negativity are associated with longer overall survival time. A cohort of 80 EAC/EGJ patients was divided according to low (TRG1–2) or high (TRG3–5) TRG scores (**a**), or nodal status and pCR (**b**) and OS time in months was visualized using a Kaplan–Meier curve. Log-rank (Mantel–Cox) tests were performed to assess median survival time differences between groups. TRG = Tumor Regression Grade, OS = Overall Survival, pCR = Pathological Complete Response.

**Figure 2 cancers-12-03356-f002:**
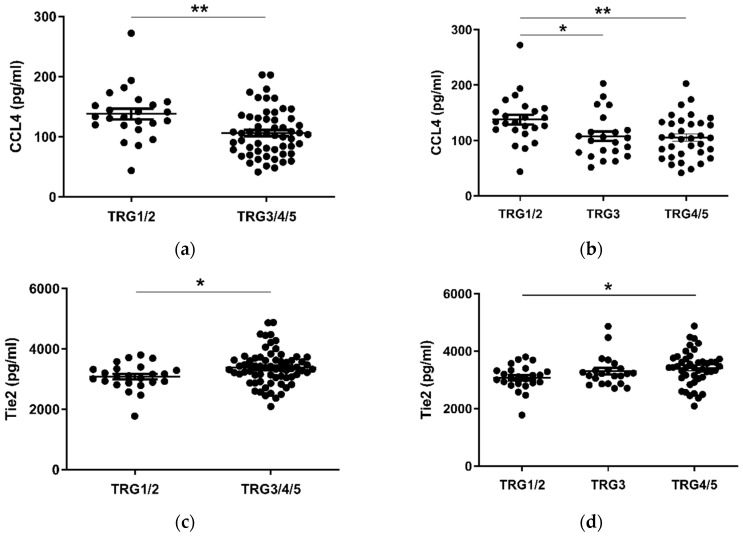
CCL4, Tie2 and CRP levels are altered in pre-treatment serum between TRG treatment response groups. Levels of cytokines were measured in serum of 80 patients using multiplex ELISA and grouped by TRG score. CCL4 levels were higher in patients with a low TRG score (**a**), particularly between the highest and lowest scored groups (**b**), whereas Tie2 (**c**) was higher in patients with high TRG scores, specifically those with the highest TRG (**d**). Levels of CRP were higher in patients with high TRG scores (**e**), though not when three tier analysis was used (**f**). Two group data sets were analyzed by Mann–Whitney test, and three group data sets were analyzed by ANOVA (Kruskal–Wallis test with Dunn’s multiple comparisons). * *p* < 0.05. ** *p* < 0.01. CRP = C Reactive Protein.

**Figure 3 cancers-12-03356-f003:**
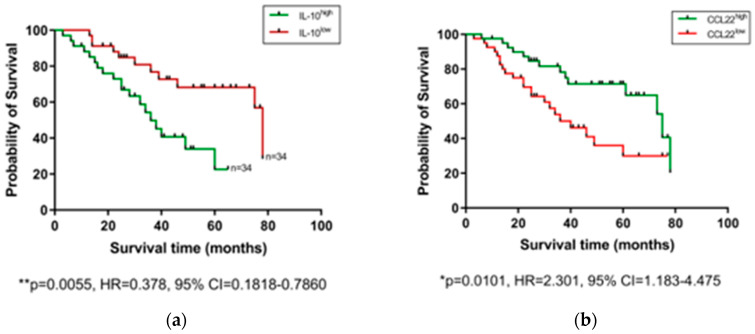
Serum IL-10, CCL22, and CCL26 levels are linked with overall survival. Groups were divided into cytokine high and low populations using the median cytokine level as a cut-off and OS was visualized using Kaplan–Meier curves. Elevated IL-10 was associated with a shorter OS time (**a**), whereas elevated levels of CCL22 (**b**) and CCL26 (**c**) were associated with longer OS time. Combining cohorts of patients with favorable cytokine conditions also showed a survival advantage, e.g., for CCL22^hi^/IL-10^lo^ (**d**) CCL26^hi^/IL-10^lo^ (**e**) and CCL22^hi^/CCL26^hi^ cohorts (**f**). Log-rank (Mantel–Cox) tests were performed to assess median survival time differences between groups and data were visualized using Kaplan–Meier curves. OS = Overall Survival, IL = Interleukin.

**Figure 4 cancers-12-03356-f004:**
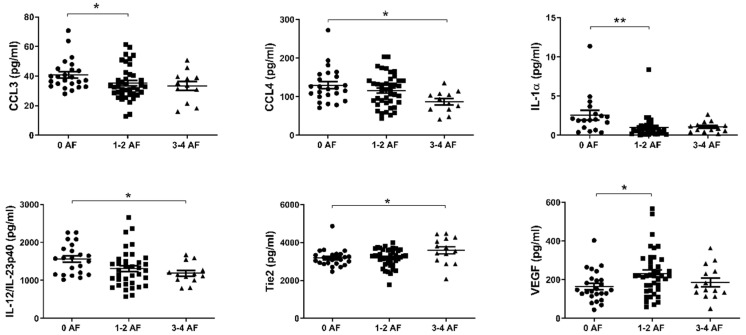
In EAC patients with no adverse features of tumor biology, CCL3, CCL4, IL-1α, and IL-12/IL-23p40 levels were higher in serum, with the contrary evident for Tie2 and VEGF. Groups were divided according to adverse features of tumor biology, with groupings of 0 adverse features, 1–2 adverse features or 3–4 adverse features. Data were analyzed by ANOVA (Kruskal–Wallis test with Dunn’s multiple comparisons). * *p* < 0.05. ** *p* < 0.01. IL = Interleukin, VEGF = Vascular Endothelial Growth Factor.

**Figure 5 cancers-12-03356-f005:**
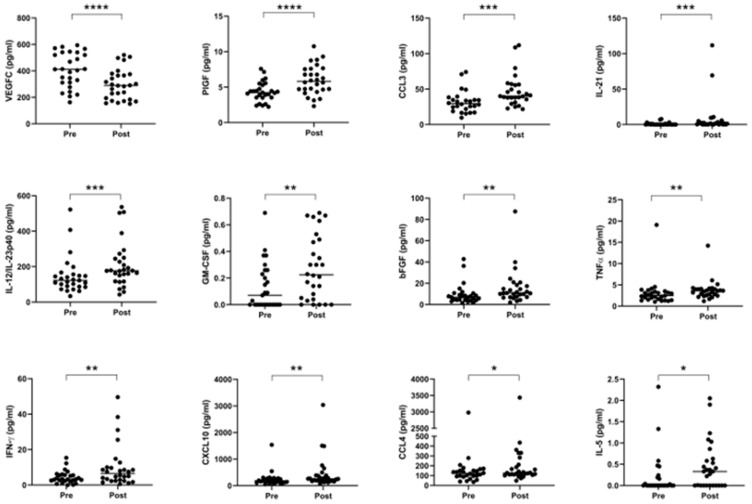
Serum pro-inflammatory cytokines are increased, and anti-angiogenic cytokines are decreased following neo-adjuvant treatment. Cytokines were measured before and after treatment in a cohort of 28 matched donors. Levels of VEGFC were lower after treatment, but other proteins were elevated, specifically PlGF, CCL3, IL-21, IL-12/IL-23p40, GM-CSF, bFGF, TNF-α, IFN-γ, CXCL10, CCL4, and IL-5. Pre- and post-treatment datasets were compared using a Wilcoxon matched pairs signed rank test, * *p* < 0.05, ** *p* < 0.01, *** *p* < 0.001, **** *p* < 0.0001. VEGF = Vascular Endothelial Growth Factor, PlGF = Placental Growth Factor, IL = Interleukin, GM-CSF = Granulocyte-Macrophage Colony-Stimulating Factor, bFGF = Basic Fibroblast Growth Factor, TNF = Tumor Necrosis Factor, IFN = Interferon.

**Figure 6 cancers-12-03356-f006:**
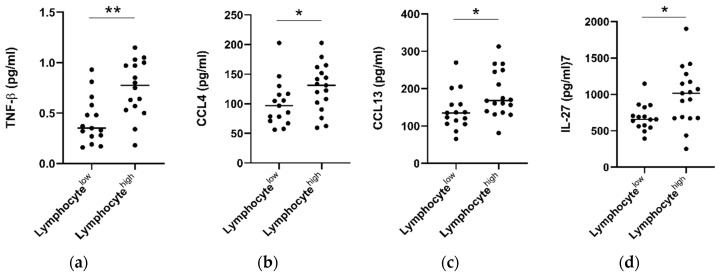
Patients with high tumor lymphocyte infiltration show elevated serum TNF-β, CCL4, CCL13, and IL-27. Tumor diagnostic biopsies were scored for lymphocytic infiltration in a cohort of n = 32 patients and divided into low (<50% infiltration, n = 15) and high (>50%, n = 17) cohorts. Pre-treatment cytokine levels were analyzed between groups and TNF-β, CCL4, CCL13, and IL-27 levels were observed to be elevated in EAC patients with high levels of lymphocytic infiltration. A Mann–Whitney test was used to compare lymphocyte low and high populations, * *p* < 0.05, ** *p* < 0.01. TNF = Tumor Necrosis Factor, IL = Interleukin.

**Figure 7 cancers-12-03356-f007:**
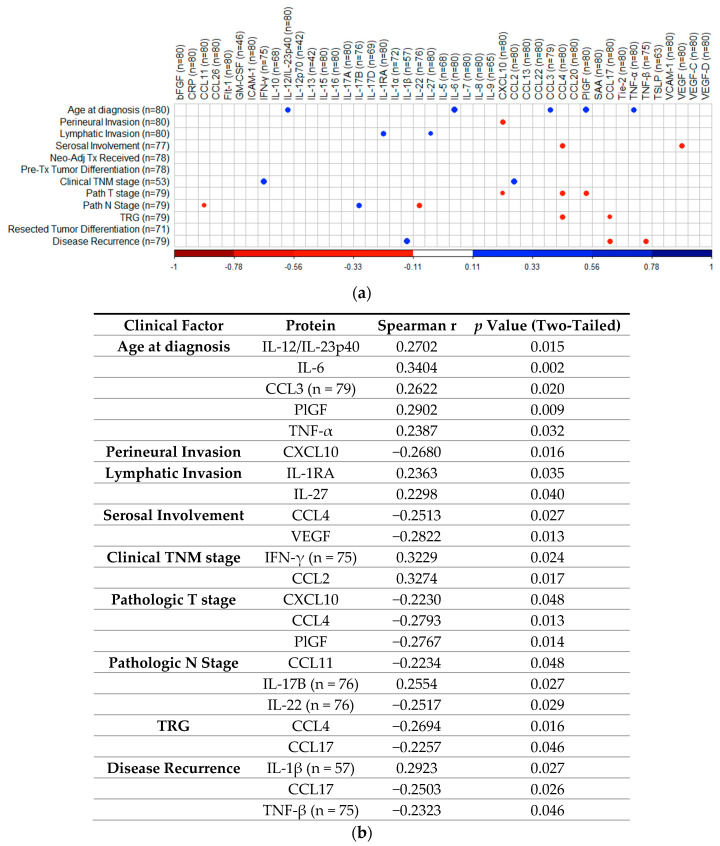
Correlation of cytokine levels with clinical parameters. (**a**) CorrPlot illustrating positive correlations identified between age at diagnosis with IL-12/IL-23p40, IL-6, CCL3, PlGF, and TNF-α, lymphatic invasion with IL-1RA and IL-27, clinical TNM stage with IFN-γ and CCL2, pathologic N stage with IL-17B and disease recurrence with IL-1β. Negative correlations also illustrated were identified between perineural invasion with CXCL10, serosal involvement with CCL4 and VEGF, pathologic T stage with CXCL10, CCL4, and PlGF, pathologic N stage with CCL11 and IL-22, TRG with CCL4 and CCL17 and disease recurrence with CCL17 and TNF-β. (**b**) Table summarizing correlation values including Spearman r and *p*-values. Cohort n = 80, unless stated otherwise. All reported *p* values were also adjusted for false discovery rate using the Holm–Bonferroni method. TNM = Tumor Node Metastasis, TRG = Tumor Regression Grade. IL = Interleukin, PlGF = Placental Growth Factor, TNF = Tumor Necrosis Factor, VEGF = Vascular Endothelial Growth Factor, IFN = Interferon.

**Table 1 cancers-12-03356-t001:** Patient demographics for pre-treatment serum ELISA analysis, n = 80.

Clinical Characteristics	
**Age in years, mean (range)**	63.9 (38–80)
**Sex (M:F)**	74:6
**Diagnosis**	
EAC	72
EGJ	7
EAC/EGJ	1
**Clinical T stage**	
T0	1
T1	0
T2	6
T3	45
T4	0
Not reported	28
**Clinical N stage**	
N0	28
N1	18
N2	7
Not reported	27
**Tumor Differentiation**	
Poor	28
Poor–Moderate	11
Moderate	33
Well	2
Not reported	6
**Pathologic T stage**	
T0	8
T1	7
T2	15
T3	48
T4	1
Not reported	1
**Pathologic N stage**	
N0	44
N1	24
N2	7
N3	5
**Tumor Differentiation (resected tissue)**	
Poor	32
Poor-Moderate	2
Moderate	31
Well	5
Not reported	10
**Neo-adjuvant treatment**	
Chemotherapy	36
Chemoradiotherapy	44
**Post-treatment Mandard TRG**	
TRG1	8
TRG2	16
TRG3	22
TRG4	21
TRG5	13
**Disease recurrence**	
Recurrence	42
No recurrence	37
Not reported	1
**Perineural invasion**	
Yes	11
No	53
Not reported	16
**Lymphovascular invasion**	
Yes	38
No	42
**Serosal involvement**	
Yes	11
No	66
Not reported	3

ELISA = Enzyme-Linked Immunosorbent Assay, EAC = Esophageal Adenocarcinoma, EGJ = Cancer of the Esophagogastric Junction, TRG = Tumor Regression Grade.

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
