# Peer review of "Linking Circulating Serum Proteins with Clinical Outcomes in Esophageal Adenocarcinoma—An Emerging Role for Chemokines"

_cancers, 2020, doi:10.3390/cancers12113356_

Round 1
Reviewer 1 Report
Dear Authors,
The study shows the analysis of immune-related proteins in the serum of patients with esophageal adenocarcinoma or gastroesophageal cancer. The protein levels are correlated with clinical and pathological parameters. Serum proteins are associated with treatment response, overall survival, adverse tumor features. This is an interesting study is, well written and and of value for the topic. I suggest minor revision. Enlish minor revisions are needed.
Minor revisions
- Fig. 3, (f) change with “CCL22hi/CCL26hi cohorts”
- line 196 ( results) 2.5, Are there 28 or 27 donors ?
- lines 274 to 277 the phrase is not demostrated by the results obtained
Reviewer 2 Report
The study by Donlon and colleagues shows the analysis of 54 immune-related proteins in the serum of 80 patients with esophageal adenocarcinoma (EAC) or gastroesophageal cancer. The protein levels are correlated with clinical and pathological parameters. Further, pre- and post-treatment serum was compared of 27 patients. A number of serum protein associations with treatment response, overall survival, adverse tumor features, and lymphocytic infiltration are described as well as differences in serum immune proteins between pre- and post-treatment time points. The study is well presented and of value for the field.
Major point
It seems that no statistical correction for multiple testing has been performed. This is a major concern. For every clinicopathological association analysis (e.g. correlation with TRG score) 54 tests have been performed. As I understand, all p-values shown are nominal p-values. It is absolutely mandatory to correct them for multiple testing, e.g. by Bonferroni correction. It is possible that many of the described associations will lose significance. In such cases, at most a non-significant trend can be reported. Each analysis, i.e. treatment response, overall survival, adverse tumor features, lymphocytic infiltration, and pre-/post-treatment, have to be corrected for 54 tests.
Minor points
- Some of the supplementary dot/box plots seem to show less than 80 samples. Is this a wrong impression or due to the resolution of the image? If some samples have been excluded for some proteins due to technical reasons, the number of successfully analyzed samples per protein should be shown in each plot and the reason for data exclusion should be mentioned in the methods. In the results section 2.7 line 237 it is mentioned that nine cytokines with fewer than 40 detectable readings were excluded from the analysis. Was that a general rule for all analyses? If yes, this should be mentioned in the methods section.
- I suggest to invert the hazard ratios (HR) so that a high level of a prognostic negative parameter results in a HR with a value >1. For example, in Fig. 3a IL-10-high results in a HR of 0.378. I suggest to change it to 1/0.378 = 2.646. This is more intuitive. This would also affect the confidence intervals. However, many studies do not reflect the direction of the measured parameter as the authors do in the present study and I leave it to the authors how to handle this.
- Legend of Fig. 3, (f) should read “CCL22hi/CCL26hi cohorts” instead of “CCL26hi/CCL26hi cohorts”
- I suggest to spell out the adverse features in the results section 2.4.
- The authors should discuss why in Fig 4 for a number of serum proteins there is a difference between 0 and 1-2 adverse features but not between 0 and 3-4 adverse features.
- In results section 2.5 line 197, the authors mention n=28 donors while the related Fig. 5 legend, 27 donors are mentioned. This has to be harmonized.
- In the discussion. lines 274 to 277 the authors say that it has been shown that TRG 1-2 vs. TRG 3-5 had a more prognostic value than TRG 1 vs TRG 2-3 vs TRG 4-5, in agreement with the present study. Have the authors performed the TRG 1 vs TRG 2-3 vs TRG 4-5 analysis?
- Line 316 uses a new acronym “OAC” which should be changed to EAC or should be introduced.
Round 2
Reviewer 2 Report
The authors have addressed all my points. The manuscript merits publication.